# Kidney Allograft Monitoring by Combining Donor-Derived Cell-Free DNA and Molecular Gene Expression: A Clinical Management Perspective

**DOI:** 10.3390/jpm13081205

**Published:** 2023-07-29

**Authors:** Asim Rizvi, Sara Faiz, Parin H. Thakkar, Syed Hussain, Ann N. Gamilla-Crudo, Michael Kueht, Muhammad A. Mujtaba

**Affiliations:** 1Department of Nephrology, Hypertension and Transplant Medicine, The University of Texas Medical Branch at Galveston, Galveston, TX 77555, USA; asrizvi@utmb.edu (A.R.); phthakka@utmb.edu (P.H.T.); sahussai@utmb.edu (S.H.); angamill@utmb.edu (A.N.G.-C.); 2Department of Pathology, Baylor College of Medicine, Houston, TX 77030, USA; sara_faiz@ymail.com; 3Department of Transplant Surgery, The University of Texas Medical Branch at Galveston, Galveston, TX 77555, USA; mlkueht@utmb.edu

**Keywords:** donor-derived cell-free DNA, Molecular Microscope Diagnostic System (MMDx^®^), rejection, antibody-mediated rejection, donor-specific antibody

## Abstract

Donor-derived cell-free DNA (dd-cfDNA) may safely assess kidney allograft rejection. Molecular Microscope (MMDx^®^) gene expression may offer increased precision to histology. This single-center retrospective study monitored kidney transplant recipients for rejection at specified time intervals by utilizing creatinine (SCr), proteinuria, donor-specific antibodies (DSAs), and dd-cfDNA. A clinically indicated biopsy sample was sent for histopathology and MMDx^®^. Patients were categorized into rejection (Rej) and non-rejection (NRej) groups, and further grouped according to antibody-mediated rejection (ABMR) subtypes. Rej and NRej groups included 52 and 37 biopsies, respectively. Median follow-up duration was 506 days. DSAs were positive in 53% and 22% of patients in both groups, respectively (*p* = 0.01). Among these groups, pre- and post-intervention median SCr, proteinuria, and dd-cfDNA at 1 month, 2 months, and at the last follow-up revealed significant difference for dd-cfDNA (all *p* = 0.01), however, no difference was found for SCr and proteinuria (*p* > 0.05). The AUC was 0.80 (95% CI: 0.69–0.91), with an optimal dd-cfDNA criterion of 2.2%. Compared to histology, MMDx^®^ was more likely to diagnose ABMR (79% vs. 100%) with either C4d positivity or negativity and/or DSA positivity or negativity. Hence, a pre- and post-intervention allograft monitoring protocol in combination with dd-cfDNA, MMDx^®^, and histology has aided in early diagnosis and timely individualized intervention.

## 1. Introduction

Allograft rejection continues to be a concern in kidney transplant recipients, as 10% of patients experience rejection in their first year after transplant [1]. The early detection and timely management of rejection using traditional markers, including serial measurements of serum creatinine, has been inadequate due to its lagging response for tissue injury. In fact, studies have repeatedly revealed serum creatinine as a poorly sensitive or specific marker of rejection [2]. Moreover, traditional and current standards of diagnosis of allograft rejection by histologic assessment of transplant kidney biopsies can have considerable inter-observer disagreement and sampling error [3,4]. An inadequate specimen is yielded in up to 15% of biopsies, hence exposing the patients to invasive procedural risk without diagnostic benefit [5]. The intricacy of traditional histology interpretation therefore reveals the need to establish novel diagnostic tools that can not only independently provide precision to the diagnosis of tissue injury and allograft rejection, but can also optimize response to rejection treatment.

Donor-derived cell-free DNA (dd-cfDNA) is a robust non-invasive molecular biomarker that has gained a substantial adoption in real-life clinical practice. It may safely and quantitatively assess tissue injury and discriminate rejection ahead of pathological findings in kidney transplant patients given its accuracy and ease of use [6]. Numerous studies have demonstrated the utility of dd-cfDNA as a clinically validated test in a broad array of contexts [2,6,7,8,9,10,11,12,13,14,15,16]. The ADMIRAL (Assessing AlloSure Dd-cfDNA, Monitoring Insights of Renal Allografts with Longitudinal Surveillance) study advocated the routine monitoring of dd-cfDNA for early detection of clinically significant graft injury and to complement histology and traditional laboratory surveillance strategies [17]. The study stated that a dd-cfDNA cut-off of >0.5% significantly correlated with clinical and subclinical rejection and was associated with increased risk of de novo DSAs [17]. Additionally, dd-cfDNA levels were found to be elevated 3 months before detectable DSA [17]. Sidgel et al., in a clinical validation study, demonstrated the ability of dd-cfDNA to discriminate active rejection from non-rejection, with a sensitivity of 88.7%, specificity of 72.6%, and area under the curve (AUC) of 0.87 using a dd-cfDNA cut-off of 1% [6]. The DART (Circulating Donor-Derived Cell-Free DNA in Blood for Diagnosing Active Rejection in Kidney Transplant Recipients) study interestingly demonstrated a strong correlation of dd-cfDNA of >1% with antibody-mediated rejection (ABMR), however, less robust association was reported for T-cell-mediated rejection (TCMR), especially early TCMR [2]. Similar findings were also reported by Huang et al. [18].

The Molecular Microscope Diagnostic System (MMDx^®^) measures mRNA transcript levels in kidney biopsy samples and applies an algorithm to score results. It uses gene expression profiling to assess disease states in a biopsy sample and can not only assess allograft injury and rejection but may offer clarity in histologically challenging situations, as well as increased precision to histology variations on kidney biopsy [19,20,21,22]. Agreement between MMDx^®^ and traditional histology was found 75–80% of the time [23]. Clinicians have reported that agreement between MMDx^®^ and clinical judgment is significantly more (87%) than histology (80%) (*p* = 0.004), and that MMDx^®^ can increase management confidence when compared to conventional assessment alone [23]. Nevertheless, the superiority of histology over MMDx^®^ was claimed in biopsies with infarcted or extensively scarred tissue and recurrent or de novo diseases [24]. The relevance of validated molecular assays in kidney transplant diagnoses was acknowledged in the 2017 Banff diagnostic classification [25]. Additionally, the more recent Banff consensus report acknowledged the limitations of histology in classification of ABMR when microvascular inflammation is present but DSAs are absent and C4d staining is negative, and states that molecular assays can help to clarify equivocal cases [22].

We utilize these state-of-the-art diagnostic tools and monitor at specified time intervals for early detection and management of kidney allograft rejection. The present study is unique as it seeks to assess the relationship of traditional laboratory markers, dd-cfDNA, histology, and MMDx^®^ for allograft rejection, along with its subtypes and subsequent tailored treatment approach in a clinical practice.

## 2. Materials and Methods

This single-center institutional review board (IRB) approved retrospective study of prospectively collected data included kidney transplant recipients at our institute who underwent clinically indicated biopsy between January 2018 to December 2021. These transplant recipients were monitored for rejection at specified time intervals by utilizing traditional laboratory markers, including serum creatinine (SCr), urine protein-to-creatinine ratio (UPCR), human leukocyte antigen donor-specific antibodies (HLA DSAs), and dd-cfDNA assay.

The clinically indicated kidney allograft biopsy was performed accordingly and a sample was sent for both histopathology and MMDx^®^. Clinical indications for biopsy included a rise in SCr of >30% above presumed baseline after ruling out reversible causes, new onset proteinuria (>1 g/day), and/or unexplained >50% elevation in dd-cfDNA above baseline. Biopsies were processed at our pathology laboratory and tissue adequacy for histology was determined by the pathologist. Tissue adequacy for MMDx^®^ was determined by the commercial laboratory to which it was shipped for analysis (Kashi Lab, Portland, Oregon, USA). Grading of rejection was performed as per the updated Banff 2017 criteria and the molecular working group [25,26]. Patients were excluded if biopsy showed evidence of BK virus nephropathy (BKVN), thrombotic microangiopathy (TMA), or recurrence of primary disease.

The SCr, UPCR, dd-cfDNA, and HLA DSAs were assessed pre-intervention at the time of biopsy and post-intervention at 1 month, 2 months, and at last follow-up. The dd-cfDNA samples were sent for commercial testing to CareDx (Brisbane, CA, USA). The exclusion criteria for dd-cfDNA assay were either <4 weeks post-transplant, pregnancy, or recipient of a multiorgan or identical twin transplant.

Patients were categorized into all-cause rejection (Rej) group, if they had evidence of rejection on histology or MMDx^®^ (either antibody-mediated [ABMR], T-cell-mediated [TCMR], or mixed), and non-rejection (NRej) group. Additionally, patients were subcategorized into mixed rejection (Mixed Rej) if they had evidence of both ABMR and TCMR on traditional histology or MMDx^®^.

Patients were subdivided into ABMR and NRej groups, and further into C4d positive ABMR (C + ABMR) and C4d negative ABMR (C-ABMR) groups. Patients with ABMR and elevated serum creatinine were managed according to our center protocol of 2 g/kg intravenous immunoglobulins (IVIG)/plasma exchange +/− Bortezomib, whereas patients with C-ABMR with stable serum creatinine were given 2 g/kg IVIG. Maintenance immunosuppression was increased in both groups.

Patients were further subdivided into ABMR with HLA DSA positivity (D + ABMR) and ABMR with HLA DSA negativity (D-ABMR) groups. Furthermore, patients were categorized into acute ABMR (ABMR) and chronic active ABMR (cABMR). Patients with cABMR were managed according to our center protocol of 2 g/kg IVIG and an increase in maintenance immunosuppression.

### Statistical Methods

Inferential statistical analysis was performed using a chi-squared test or Fisher’s exact test for categorical variables and Student’s *t*-test or Mann–Whitney’s U test for normalized/nonnormalized variables as indicated. Receiver operating characteristic (ROC) curves were created for dd-cfDNA scores and rejection diagnoses by histology or MMDx^®^. The optimal cut-off to provide best sensitivity and specificity was derived based upon the Youden Index. The *p*-values were calculated using Mann–Whitney’s U test and a *p*-value of <0.05 was considered statistically significant.

## 3. Results

The characteristics of patients included in the study are listed in Table 1. A total of 101 biopsies from 89 patients (53 ± 13 years) were included. The biopsies suggestive of BKVN, TMA, and recurrence of primary disease were excluded, and a total of 89 biopsies were finally included in the study (Figure 1). The median (IQR) follow-up duration was 506 (340, 660) days. One year kidney allograft survival for these patients was 98%.

### 3.1. All-Cause Rejection (Rej) vs. Non-Rejection (NRej) Groups

The Rej group included 52 biopsies from 49 patients (50 ± 13 years) and the NRej group included 37 biopsies from 36 patients (56 ± 12 years). The HLA DSAs were positive in 26 (53%) patients in the Rej group and eight (22%) patients in the NRej group (*p* = 0.01). In the Rej group, pre- and post-intervention median (IQR) SCr, UPCR, and dd-cfDNA were compared to NRej group at specified time intervals and depicted in Figure 2A. When comparing the Rej and NRej groups, a significant difference was found in pre-intervention median dd-cfDNA (2.25 vs. 0.44, *p* = 0.01), whereas median SCr (1.77 vs. 2.50) and UPCR (0.29 vs. 0.53) did not achieve statistical significance with *p*-values of 0.36 and 0.38, respectively. Among the Rej group, histology was found to be negative for rejection in six biopsies (12%) and MMDx^®^ was negative in seven biopsies (14%). The receiver operating characteristics curve (ROC) curve for dd-cfDNA and rejection diagnoses is shown in Figure 3A. Area under the curve (AUC) was 0.80 (95% CI: 0.69–0.91) with an optimal dd-cfDNA criterion of 2.2%.

### 3.2. Antibody-Mediated Rejection (ABMR) vs. Non-Rejection (NRej) Groups

The ABMR group included 28 biopsies from 28 patients (50 ± 13 years) and the NRej group included 37 biopsies from 36 patients (56 ± 12 years). The HLA DSAs were positive in 15 (54%) patients in the ABMR group and eight (22%) patients in the NRej group (*p* = 0.02). In the ABMR group, pre- and post-intervention median (IQR) SCr, UPCR, and dd-cfDNA were compared to the NRej group at specified time intervals and are shown in Figure 2B. When comparing the ABMR and NRej groups, a significant difference was found in pre-intervention median dd-cfDNA (2.45 vs. 0.44, *p* = 0.01), whereas SCr (1.45 vs. 2.50) and UPCR (0.82 vs. 0.53) did not achieve significance with *p*-values of 0.71 and 0.85, respectively. Among the ABMR group, histology was found to be negative for ABMR in six biopsies (21%), while MMDx^®^ was positive for ABMR in 100% of biopsies. The receiver operating characteristics curve (ROC) curve for dd-cfDNA and ABMR diagnoses is shown in Figure 3B. The area under the curve (AUC) was 0.84 (95% CI: 0.73–0.96) with an optimal dd-cfDNA criterion of 2.2%.

### 3.3. Mixed Rejection (Mixed Rej) vs. Non-Rejection (NRej) Groups

The Mixed Rej group included 17 biopsies from 17 patients (50 ± 15 years) and the NRej group included 37 biopsies from 36 patients (56 ± 12 years). The HLA DSAs were positive in 12 (71%) patients in the Mixed Rej group and eight (22%) patients in the NRej group. In the Mixed Rej group, pre- and post-intervention median (IQR) SCr, UPCR, and dd-cfDNA were compared to the NRej group at specified time intervals and are represented in Figure 2C. When comparing the Mixed Rej and NRej groups, a significant difference was noted for pre-intervention median dd-cfDNA (1.30 vs. 0.44, *p* = 0.01), whereas median SCr (2.50 vs. 2.50) and UPCR (0.29 vs. 0.53) did not achieve significance with *p*-values of 0.41 and 0.37, respectively. Among the Mixed Rej group, histology revealed grade 1A, 1B, and 2A TCMR in one (6%), 11 (65%), and five (29%) patients, respectively. Traditional histology was found to be negative for ABMR in 13 biopsies (77%), whereas MMDx^®^ was positive for ABMR in 100% of biopsies. The receiver operating characteristics curve (ROC) curve for dd-cfDNA and mixed rejection diagnoses is shown in Figure 3C. The area under the curve (AUC) was 0.79 (95%CI: 0.62–0.95) with an optimal dd-cfDNA criterion of 0.70%.

### 3.4. C4d Positive ABMR (C + ABMR) vs. C4d Negative ABMR (C-ABMR) Groups

The C + ABMR group included 11 biopsies from 11 patients (49 ± 14 years) and the C-ABMR group included 17 biopsies from 17 patients (50 ± 13 years). The HLA DSAs were positive in six (55%) patients in the C + ABMR group and four (24%) patients in the C-ABMR group (*p* > 0.05). Among the C + ABMR and C-ABMR groups, no significant differences were observed in pre-intervention median dd-cfDNA (2.80 vs. 2.30, *p* = 0.38), SCr (1.78 vs. 2.50, *p* = 0.17), and UPCR (0.25 vs. 0.53, *p* = 0.72) (Appendix A). Among the C + ABMR and C-ABMR groups, histology was negative for ABMR in two (18%) and four biopsies (24%), respectively. The MMDx^®^ was positive for ABMR in 100% of biopsies in both groups.

### 3.5. ABMR with HLA DSA Positivity (D + ABMR) vs. ABMR with HLA DSA Negativity (D-ABMR) Groups

The D + ABMR group included 15 biopsies from 15 patients (52 ± 12 years) and the D-ABMR group included 13 biopsies from 13 patients (47 ± 14 years). Among the D + ABMR and D-ABMR groups, no significant differences were noted for pre-intervention dd-cfDNA (2.50 vs. 2.40, *p* = 0.73), SCr (1.36 vs. 1.54, *p* = 0.69), and UPCR (1.18 vs. 0.29, *p* = 0.09) (Appendix A). Among the D + ABMR and D-ABMR groups, histology was negative for ABMR in two (13%) and four (31%) biopsies, respectively. The MMDx^®^ was positive for ABMR in 100% of biopsies in both groups.

### 3.6. Acute Antibody-Mediated Rejection (ABMR) vs. Chronic Active Antibody-Mediated Rejection (cABMR)

The ABMR group included 18 biopsies from 18 patients (49 ± 14 years), the cABMR group included 10 biopsies from 10 patients (50 ± 12 years), and the NRej group included 37 biopsies from 36 patients (56 ± 12 years). The HLA DSAs were positive in 39%, 70%, and 22% of patients in the ABMR, cABMR, and NRej groups, respectively. Among the ABMR and cABMR groups, no significant differences were observed in pre-intervention median dd-cfDNA (2.50 vs. 2.50, *p* = 0.59) and SCr (1.30 vs. 1.74, *p* = 0.31), however, a significant difference was noted for pre-intervention median UPCR (0.25 vs. 1.77, *p* = 0.03) (Figure 4A). Additionally, when comparing the cABMR and NRej groups, a significant difference was noted for pre-intervention median (IQR) dd-cfDNA (*p* < 0.01), however, no significant differences were found for pre-intervention median (IQR) SCr and UPCR (all *p* > 0.05) (Figure 4B). The receiver operating characteristics curve (ROC) curve for dd-cfDNA and cABMR diagnoses is shown in Figure 4C. The area under the curve (AUC) was 0.93 (95% CI: 0.69–0.91) with an optimal dd-cfDNA criterion of 2.1%. Among the cABMR group, a significant difference was observed in the pre- and post-intervention (at last follow-up) median (IQR) dd-cfDNA [2.50 (2.30, 3.61) vs. 0.90 (0.40, 1.35), *p* = 0.01], however, SCr and UPCR did not achieve significance with *p*-values of 0.36 and 0.35, respectively.

### 3.7. Pre-Intervention and Post-Intervention Comparison

Among the all-cause rejection group, a significant difference was observed in the pre- and post-intervention (at last follow-up) median (IQR) dd-cfDNA [2.25 (0.69, 3.80) vs. 1.02 (0.29, 2.12), *p* = 0.0] with relative change value (RCV) of −55%. However, the SCr [1.77 (1.32, 3.02) vs. 1.73 (1.28, 2.93)] and UPCR [0.29 (0.17, 1.13) vs. 0.34 (0.10, 1.17)] did not achieve significance with *p*-values of 0.84 and 0.41, respectively.

## 4. Discussion

This study has assessed the utility of available diagnostic tools, including dd-cfDNA and MMDx^®^, in addition to traditional laboratory markers and histology for early detection and timely management of kidney allograft rejection. Most notably, our study includes allograft rejection subtypes and herein reports on the response to intervention, hence expanding upon previously published data [2,5,26].

To this end, while comparing the possibility of rejection diagnoses by published dd-cfDNA criterion of ≥1%, our study yielded AUC to be close to 0.80 with a specificity of 79% for a dd-cfDNA cut-off of 1% among all-cause rejection, antibody-mediated rejection, and mixed rejection groups. The relatively low sensitivities for the dd-cfDNA cut-off of 1% among these groups are at 62%, 71%, and 62%, respectively, and indicate that dd-cfDNA is susceptible to false negative results. However, a higher sensitivity of more than 80% was established for a dd-cfDNA cut-off of 0.5% among these groups. These findings therefore advocate for a dd-cfDNA cut-off of 0.5% to be more sensitive to ‘rule-out’ and a cut-off of 1% to be more specific to ‘rule-in’ all-cause rejection and its subtypes’ diagnoses.

An interesting observation of the current study is that dd-cfDNA potentially identified cases of histologic antibody-mediated rejection without detectable HLA DSAs, where antibody-mediated rejection could likely be mediated by non-HLA DSAs, such as antibodies, to the type 1 angiotensin receptor (anti-AT1R) [27,28]. Furthermore, dd-cfDNA not only down-trended significantly and responded well to our center protocol of tailored treatment for allograft rejection, as noted above, but levels for dd-cfDNA were maintained throughout the median follow-up duration of more than 500 days, thereby improving allograft survival.

In addition, discrepancies were found between histology and MMDx^®^ diagnoses as noted above. When compared to histology, MMDx^®^ was more likely to diagnose antibody-mediated rejection (79% vs. 100%) with either C4d positivity or negativity and/or HLA DSA positivity or negativity. However, both traditional histology and MMDx^®^ showed agreement in diagnosing T-cell-mediated rejection among patients included in our study. On the other hand, histology was more likely to diagnose all-cause rejection when compared to MMDx^®^ (88% vs. 86%). Hence, current study findings support utilizing a combination of traditional histology with molecular gene expression among patients and a dd-cfDNA cut-off of ≥1%, given its high specificity to ‘rule-in’ allograft rejection and its diagnosis of its subtypes. It is also worth mentioning here that dd-cfDNA, though very helpful for rejection diagnoses, was not able to differentiate between various histological subtypes of antibody-mediated rejection, or antibody-mediated versus mixed rejection.

Another important observation made in this study was that T-cell-mediated rejection on traditional histology was also reported to have a mild antibody-mediated component on MMDx^®^, and these patients were categorized under mixed rejection. There was only one patient with pure T-cell-mediated rejection reported both on traditional histology and MMDx^®^ testing, and this therefore limits our study’s ability to comment on dd-cfDNA’s capability to distinguish between antibody-mediated and pure T-cell-mediated rejection.

In our early experience of dd-cfDNA, we had two patients with elevated levels but with normal renal function based on serum creatinine and proteinuria as well as absence of HLA DSAs, who developed biopsy-proven antibody-mediated rejection within 2 months of elevated dd-cfDNA. This prompted us to include an elevated dd-cfDNA level of >1% and/or >50% increase from baseline dd-cfDNA as one of the criteria for kidney biopsy in an appropriate clinical setting. The next question was the management for such patients if allograft biopsy showed changes consistent with humoral response despite stable serum creatinine. We approached these patients with increasing maintenance immunosuppression and IVIG and observed decline in dd-cfDNA levels. There were two patients who underwent repeat allograft biopsy after this management approach and were found to have a resolution of humoral response on traditional histology and MMDx^®^.

Further still, our study reported allograft monitoring, although the follow-up duration was for 500 days at that point. With our patient-centric approach, we did not observe a higher incidence of infectious complications.

This study had limitations, including single-center retrospective study design with a relatively small sample size and shorter follow-up duration. Moreover, patients were included who had pre-intervention dd-cDNA performed within 15 days of the allograft biopsy, which might have impacted the study findings, as dd-cfDNA has a much shorter half-life. However, in real clinical practice, the process of obtaining biopsy and dd-cfDNA on the same day is, practically, not feasible due to patients’ preferences and scheduling limitations. Moreover, caution should be taken not to rely on thresholds for rejection interpretation, as the specific threshold may vary from center to center and patient to patient. The current study was also limited due to lack of non-HLA antibody testing, which has its own challenges.

As a single-center study, there is more consistency in immunosuppression, monitoring protocols, biopsy read, intervention protocol, and follow-up. Our study highlights the important role of dd-cfDNA in day-to-day transplant patient care, which can help with allograft monitoring, timely intervention, and response to therapy. Similarly, MMDx^®^ in our experience is complementary to traditional histology. We emphasize here that dd-cfDNA testing frequency and threshold cut-offs should be individualized based on patient rejection risk profile. Future studies are warranted to confirm and expand on our findings.

## 5. Conclusions

The utilization of allograft rejection monitoring and post-intervention protocol with inclusion of dd-cfDNA and addition of MMDx^®^ to kidney biopsy has aided in early recognition, timely intervention, and individualized treatment in our practice. Inclusion of these modalities may help in improving allograft survival.

## Figures and Tables

**Figure 1 jpm-13-01205-f001:**
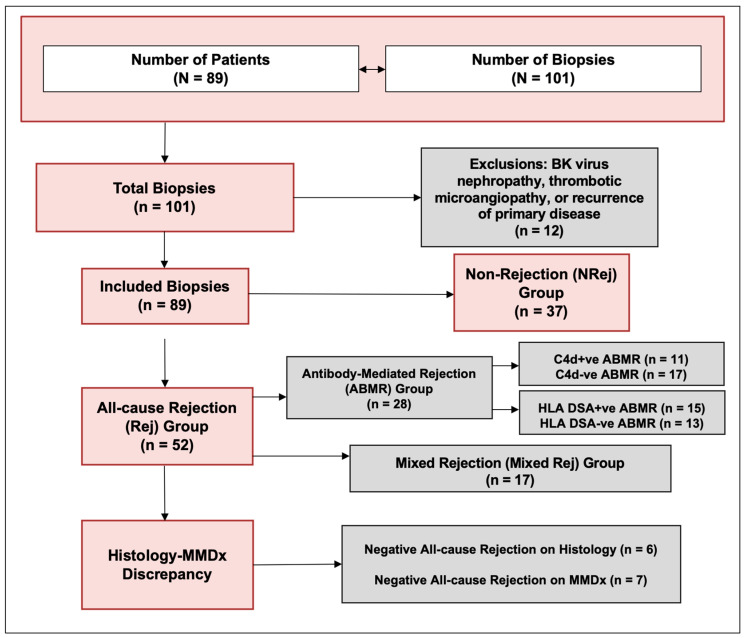
Study flow diagram.

**Figure 2 jpm-13-01205-f002:**
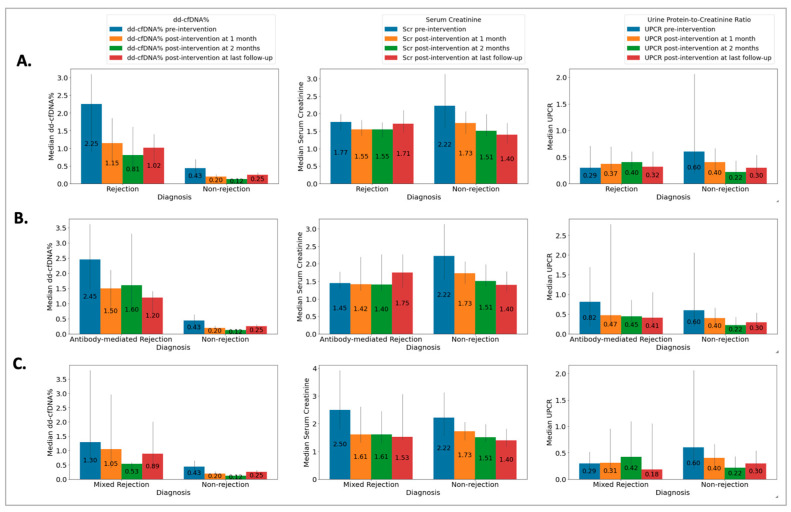
Relationship among (**A**) rejection (Rej) vs. non-rejection (NRej) groups [significant difference was found in pre-intervention median (IQR) donor-derived cell-free DNA (*p* = 0.01), whereas median (IQR) serum creatinine and urine protein-to-creatinine ratio did not achieve statistical significance with *p*-values of 0.36 and 0.38, respectively]; (**B**) antibody-mediated rejection (ABMR) vs. non-rejection (NRej) groups [significant difference was noted in pre-intervention median (IQR) donor-derived cell-free DNA (*p* = 0.01), however, serum creatinine and urine protein-to-creatinine ratio did not achieve significance with *p*-values of 0.71 and 0.85, respectively]; and (**C**) mixed rejection (Mixed Rej) vs. non-rejection (NRej) groups [significant difference was found in pre-intervention median (IQR) donor-derived cell-free DNA (*p* = 0.01), whereas median (IQR) serum creatinine and urine protein-to-creatinine ratio did not achieve significance with *p*-values of 0.41 and 0.37, respectively].

**Figure 3 jpm-13-01205-f003:**
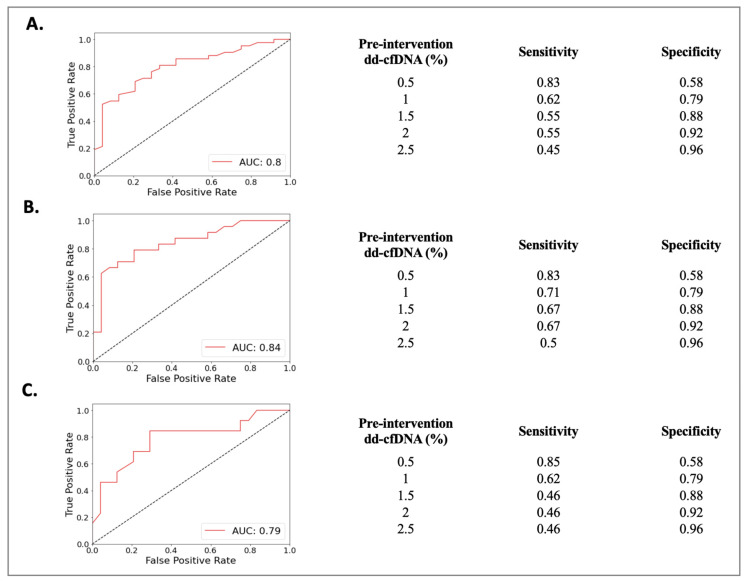
Receiver operating characteristics curve (ROC) curve for dd-cfDNA among (**A**) rejection (Rej) vs. non-rejection (NRej) groups, (**B**) antibody-mediated rejection (ABMR) vs. non-rejection (NRej) groups, and (**C**) mixed rejection (Mixed Rej) vs. non-rejection (NRej) groups.

**Figure 4 jpm-13-01205-f004:**
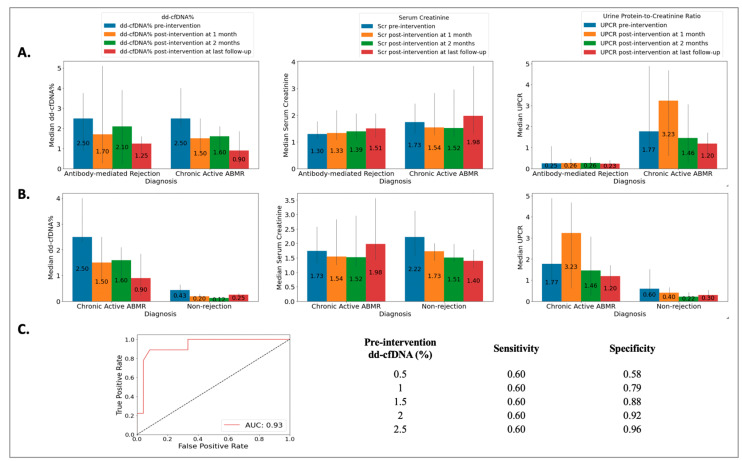
Relationship among (**A**) acute antibody-mediated rejection (ABMR) vs. chronic active antibody-mediated rejection (cABMR) groups [no significant difference was observed in pre-intervention median (IQR) donor-derived cell-free DNA (*p* = 0.59) and serum creatinine (*p* = 0.31), however, significant difference was found for pre-intervention median (IQR) urine protein-to-creatinine ratio with a *p*-value of 0.03]; (**B**) chronic active antibody-mediated rejection (cABMR) vs. non-rejection (NRej) groups [a significant difference was found in pre-intervention median (IQR) donor-derived cell-free DNA (*p* < 0.01), whereas median (IQR) serum creatinine and urine protein-to-creatinine ratio did not achieve significance (all *p* > 0.05)]; and (**C**) receiver operating characteristics curve (ROC) for donor-derived cell-free DNA (dd-cfDNA) among chronic active antibody-mediated rejection (cABMR) vs. non-rejection (NRej) groups.

**Table 1 jpm-13-01205-t001:** Patient characteristics.

	All-Cause Rejection	Antibody-Mediated Rejection	Mixed Rejection	Non-Rejection
Number of Patients (*n*)	49	28	17	36
Number of Biopsies (*n*)	52	28	17	37
Age, Years (Mean ± SD)	50 ± 13	50 ± 13	50 ± 15	56 ± 12
HLA DSA Positivity at the Time of Biopsy (*n*, %)	26, 53	14, 50	12, 71	8, 22
C4d Positivity (*n*, %)	17, 33	11, 39	5, 29	3, 8
Pre-Intervention dd-cfDNA Score, Median (IQR)	2.25 (0.69, 3.80)	2.45 (0.90, 3.72)	1.30 (0.73, 3.80)	0.44 (0.24, 0.76)
Histologic Diagnosis (*n*, %)	46, 88	22, 79	4, 23	37, 100
MMDx^®^ Diagnosis (*n*, %)	45, 87	28, 100	17, 100	37, 100
**Antibody-Mediated Rejection Subtypes**				
	C4d Positive ABMR	C4d Negative ABMR	ABMR with HLA DSA Positivity	ABMR with HLA DSA Negativity
Number of Patients (*n*)	11	17	15	13
Number of Biopsies (*n*)	11	17	15	13
Age, Years (Mean ± SD)	49 ± 14	50 ± 13	52 ± 12	47 ± 14
HLA DSA Positivity at the Time of Biopsy (*n*, %)	6, 55	4, 24	15, 100	0, 0
C4d Positivity (*n*, %)	11, 100	0, 0	7, 47	4, 31
Pre-Intervention dd-cfDNA Score, Median (IQR)	2.80 (1.90, 4.65)	2.30 (0.88, 3.61)	2.50 (1.09, 3.81)	2.40 (0.90, 3.70)
Histologic Diagnosis (*n*, %)	9, 82	13, 76	13, 87	9, 69
MMDx^®^ Diagnosis (*n*, %)	11, 100	17, 100	15, 100	13, 100

SD, standard deviation; IQR, interquartile range; dd-cfDNA, donor-derived cell-free DNA; HLA DSA, HLA donor-specific antibodies; MMDx^®^, Molecular Microscope Diagnostic System; ABMR, antibody-mediated rejection.

## Data Availability

The data presented in this study are available on request from the corresponding author. The data are not publicly available due to restrictions in privacy.

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
