# Peer review of "Kidney Allograft Monitoring by Combining Donor-Derived Cell-Free DNA and Molecular Gene Expression: A Clinical Management Perspective"

_jpm, 2023, doi:10.3390/jpm13081205_

Round 1

Reviewer 1 Report

This is a comprehensive report on the introduction in the clinical setting of a new monitoring tool, cf-ddDNA for early detection of rejection in addition to the use of a more accurate molecular technology for diagnosis of rejection in allograft biopsies. 

I have just minor comments: 

1.You mention 500 days of follow up can you provide the range of that follow up.

2. As 1-year sCr. is a predictor of long-term graft survival can you provide the levels for patients in the different groups.

There is no information on graft and patient survival which may not be the main focus of the paper, but it should be included.

Why you excluded occurrence of rejection early on during the first 4 weeks after transplant, since it is known that almost 50% of ACR and ABMR occur during that period.

While comparing rejection to no rejection you repeat the report on sf-ddDNA, sCr , UPCR levels in the no- rejection group three times.  Why not gathering all rejection groups to one paragraph showing that only sf-ddDNA was significantly higher in rejection vs. all other 3 groups. 

Lastly, since the groups were small you couldn't find significant differences in sCr and UPSC between the groups. Looking at the last follow up it seems that there was a trend towards higher sCr in the ABMR C+ compared to ABMR C-. Can you elaborate on that. 

Author Response

Thank you for the comments. Please see the response as below.

1.You mention 500 days of follow up can you provide the range of that follow up.( Line 139-140), we have provided median follow up duration 506 ( 340-660 days)

  1. As 1-year sCr. is a predictor of long-term graft survival can you provide the levels for patients in the different groups. Figure 2: Provides median dd-cfDNA, median Serum Creatinine, and UPC ratio among rejection and non-rejection groups.

There is no information on graft and patient survival which may not be the main focus of the paper, but it should be included. For this cohort of patients, 1 year graft survival was 97.7% (added line 141)

3- Why you excluded occurrence of rejection early on during the first 4 weeks after transplant, since it is known that almost 50% of ACR and ABMR occur during that period.

We use two cell free DNA level at least two weeks apart to define baseline. First test is performed after two weeks of transplant per manufacturer recommendation. At our center- Acute rejection rate <30 days are less than 5%.  

4. While comparing rejection to no rejection you repeat the report on sf-ddDNA, sCr , UPCR levels in the no- rejection group three times.  Why not gathering all rejection groups to one paragraph showing that only sf-ddDNA was significantly higher in rejection vs. all other 3 groups. 

 We compared diff types of rejection with non-rejection, no doubt there is redundancy, however it provides better visual comparison to reader in our view (Figure 2)

5- Lastly, since the groups were small you couldn't find significant differences in sCr and UPSC between the groups. Looking at the last follow up it seems that there was a trend towards higher sCr in the ABMR C+ compared to ABMR C-. Can you elaborate on that. (Finding is interesting but did not achieve statistical significance, it needs to be tested in a larger cohort. In our view C+ABMR may represent a severe or advanced AMR, and its well documented to be associated with shorter intermediate graft survival).  

Reviewer 2 Report

Rizvi et al. present a clinical experience study with 89 patients receiving clinically indicated biopsy. Testing biopsy samples with both histology and Molecular Microscope diagnostic testing.  Patients were also monitored with serum creatinine, DSA, proteinuria and donor-derived cell-free DNA, and Patients were followed up, on average, for about a year and a half.  These analytes were measured both prior to biopsy, and also after the biopsy at one month, two months, and last follow-up.

Some interesting findings include:

(1) dd-cfDNA was significantly higher in the Rej group, compared to the NRej group, while SCr and UPCR were not significantly elevated.  This finding held true when the Rej group was restricted to only biopsies showing ABMR or Mixed. 

(2) DSA was positive in only 54% of the ABMR biopsies, and that dd-cfDNA was not significantly different in C4D+ and C4D ABMR cases.

(3) Anecdotally, several patients with an elevated dd-cfDNA but normal renal function later progressed to rejection, suggesting that dd-cfDNA is a leading indicator of rejection.

The manuscript is well written, covers an important topic, and should be published.

Two comments:

1) One notable difference in the methods, as compared to most other related papers: patients were categorized into all-cause rejection group if they had evidence of rejection on histology or MMDx. In comparison, most manuscripts only use one method (histology or MMDx) to classify patients as rejecting.  The decision to use both methods for biopsy analysis, and include samples that are considered rejection by either method would be expected to result in lower sensitivities and higher specificities as compared to previous clinical validation studies.  (Figure 3) Please note this in the discussion.

2) In Figure 2 / Section 3.7, there seems to be a larger drop in dd-cfDNA after treatment in the Rej group, than in the NRej group. Is this true?  When looking at SCr in these groups, it appears that there was no drop in the Rej group after treatment, but there was a downward trend in the NRej group.  This is curious.  This could be explained if the NRej biopsies were triggered, predominantly by elevated SCr, and that the downward trend was just regression to the mean.  Is this true? 

It would be interesting to look at what proportion of for-cause biopsies in the Rej and NRej groups were triggered be elevated SCr/UPCR vs. dd-cfDNA, and if the trends in Figure 2 are similar when stratified into those biopsies triggered by SCr/UPCR vs. dd-cfDNA.  The numbers in this cohort may not be large enough to make any conclusions.

Author Response

Thank you for the review and feedback. Please see responses below.

Reviewer 2

1) One notable difference in the methods, as compared to most other related papers: patients were categorized into all-cause rejection group if they had evidence of rejection on histology or MMDx. In comparison, most manuscripts only use one method (histology or MMDx) to classify patients as rejecting.  The decision to use both methods for biopsy analysis and include samples that are considered rejection by either method would be expected to result in lower sensitivities and higher specificities as compared to previous clinical validation studies.  (Figure 3) Please note this in the discussion . Comment updated in introduction section ( studies demonstrated that certain molecular classifiers improve diagnosis of ABMR beyond what is possible with histology, C4d, and detection of donor-specific antibodies (DSAs) and that both C4d and validated molecular assays can serve as potential alternatives and/or complements to DSAs in the diagnosis of ABMR. The Banff 2017 ABMR criteria are thus updated to include Molecular assays- (Please see line 75-76 and 82 and 85)

2) In Figure 2 / Section 3.7, there seems to be a larger drop in dd-cfDNA after treatment in the Rej group, than in the NRej group. Is this true?  When looking at SCr in these groups, it appears that there was no drop in the Rej group after treatment, but there was a downward trend in the NRej group.  This is curious.  This could be explained if the NRej biopsies were triggered, predominantly by elevated SCr, and that the downward trend was just regression to the mean.  Is this true? Indication of biopsy was explained in methodology section. Patient underwent biopsy for elevation in donor derived cell free DNA, and or serum creatinine. Rejection diagnosis is biopsy results based. In Rejection patients there is immediate cause and immediate treatment, non-rejection AKI causes can be different, but most biopsies showed ATN findings, and were switched to alternate immunosuppression or treated for AIN, as pointed by reviewer appears to be regression to mean.

It would be interesting to look at what proportion of for-cause biopsies in the Rej and NRej groups were triggered be elevated sCr/UPCR vs. dd-cfDNA, and if the trends in Figure 2 are similar when stratified into those biopsies triggered by sCr/UPCR vs. dd-cfDNA.  The numbers in this cohort may not be large enough to make any conclusions.

35 % of biopsies were performed with concomitant elevation in serum creatinine and cell free DNA- 90% of biopsy in this cohort were positive for rejection. A large multicenter trial is needed to further these findings.